# Redox Imbalance and Biochemical Changes in Cancer by Probing Redox-Sensitive Mitochondrial Cytochromes in Label-Free Visible Resonance Raman Imaging

**DOI:** 10.3390/cancers13050960

**Published:** 2021-02-25

**Authors:** Halina Abramczyk, Beata Brozek-Pluska, Monika Kopec, Jakub Surmacki, Maciej Błaszczyk, Maciej Radek

**Affiliations:** 1Laboratory of Laser Molecular Spectroscopy, Institute of Applied Radiation Chemistry, Lodz University of Technology, Wroblewskiego 15, 93-590 Lodz, Poland; beata.brozek-pluska@p.lodz.pl (B.B.-P.); monika.kopec@p.lodz.pl (M.K.); jakub.surmacki@p.lodz.pl (J.S.); 2Department of Neurosurgery, Spine and Peripheral Nerve Surgery, University Hospital WAM-CSW, Medical University of Lodz, Zeromskiego 113, 91-647 Lodz, Poland; maciej.blaszczyk@umed.lodz.pl (M.B.); maciej.radek@umed.lodz.pl (M.R.)

**Keywords:** redox state, optical biopsy, Raman spectroscopy and imaging, brain and breast cancer, cytochrome *c*

## Abstract

**Simple Summary:**

Gliomas comprise around 30% of human brain tumors, while invasive ductal carcinoma (IDC) comprises around 80% of human breast cancers. The aim of our study was to show that cancerogenesis affects the redox status of mitochondrial cytochromes, which can be tracked by using Raman spectroscopy and imaging. Our results confirmed that human breast cancer and brain tumor demonstrate a redox imbalance compared to normal tissues. We have shown the correlation between the intensity of cytochromes Raman bands at 750, 1126, 1337 and 1584 cm^−1^ and malignancy grade for brain and breast cancers.

**Abstract:**

To monitor redox state changes and biological mechanisms occurring in mitochondrial cytochromes in cancers improving methods are required. We used Raman spectroscopy and Raman imaging to monitor changes in the redox state of the mitochondrial cytochromes in ex vivo human brain and breast tissues at 532 nm, 633 nm, 785 nm. We identified the oncogenic processes that characterize human infiltrating ductal carcinoma (IDC) and human brain tumors: gliomas; astrocytoma and medulloblastoma based on the quantification of cytochrome redox status by exploiting the resonance-enhancement effect of Raman scattering. We visualized localization of cytochromes by Raman imaging in the breast and brain tissues and analyzed cytochrome *c* vibrations at 750, 1126, 1337 and 1584 cm^−1^ as a function of malignancy grade. We found that the concentration of reduced cytochrome *c* becomes abnormally high in human brain tumors and breast cancers and correlates with the grade of cancer. We showed that Raman imaging provides additional insight into the biology of astrocytomas and breast ductal invasive cancer, which can be used for noninvasive grading, differential diagnosis.

## 1. Introduction

Invasive ductal carcinoma (IDC), also known as infiltrating ductal carcinoma, and gliomas are the most common types of breast cancer and brain tumors. IDCs comprises about 80 percent of all breast cancer diagnoses, gliomas comprise about 30 percent of all brain tumors and central nervous system, astrocytomas are the most common gliomas [1,2,3]. Many cancers, including ductal cancers and gliomas have long been thought to primarily metabolize glucose for energy production, a phenomenon known as the Warburg effect, which addresses the metabolic shift of most cancer cells that prefer ATP generation through enhanced glycolysis followed by lactic acid fermentation in cytosol even in the presence of oxygen instead of ATP generation through oxidative phosphorylation in mitochondrial respiration. Presently, an increasing number of reports have initiated a discussion about metabolic regulation in cancers [4,5,6] showing that metabolic adaptation in tumors extends beyond the Warburg effect. Indeed, it was discovered that in MCF-7 breast cancer cells that total ATP turnover was 80% oxidative and 20% glycolytic [5]. This hypothesis was also tested in primary-cultured human glioblastoma cells and it was found that cells were highly oxidative and largely unaffected by treatment with glucose or inhibitors of glycolysis [6]. Thus, it appears that oxidative phosphorylation can co-exist with aerobic glycolysis and lactate release.

It is becoming clear that changes in metabolism during cancer development are governed by a balance between the need of the cell for energy supply with its equally important need for macromolecular building blocks and maintenance of redox balance. Regarding macromolecular building blocks the role of fatty acids as critical bio-energetic substrates within glioma cells [6,7,8,9,10,11,12,13] and breast cancer cells [10] has been recognized. The redox balance depends on a large extend on mitochondrial functionality. To address the question of mitochondrial functionality in electron transfer chain mitochondrial enzymes expression and activity have been studied [14,15,16]. The findings of different groups are somewhat conflicted regarding a possible impairment of the respiratory chain in gliomas. Early studies on glioma cell rat xenografts identified lower cytochrome *c* oxidase (COX, Complex IV) and SDH (Complex II) enzyme expression in more hypoxic areas of the tumor. More recently, one group observed significantly lower Complex II-IV activity in anaplastic astrocytomas and lower Complex I-IV activity in glioblastomas compared with normal brain tissue, using dissociated cells from freshly frozen human tumors [14]. Another group analyzed human glioma tissue samples by mass spectrometry and observed lower expression of some Complex I subunits but higher levels of many oxidative enzymes including catalase [15].

The cytochrome family of heme-containing proteins plays a critical role in the mitochondrial mechanism of cell respiration as an electron carrier in the electron transfer chain in mechanism of oxidative phosphorylation. They are also important in intercellular cell signaling, apoptosis, and metabolizing polyunsaturated fatty acids. A recent study has shown that it can also act as an antioxidative enzyme [17,18,19,20,21]. Cytochromes are classified based on their lowest electronic energy absorption band in their reduced state: cytochrome P450 (450 nm), cytochrome *c* (550 nm), cytochromes *b* (≈565 nm), cytochromes *a* (605 nm). The family of cytochromes are localized in the third complex in the electron transport chain, also known as complex III or Coenzyme Q - Cyt *c* reductase, sometimes called the cytochrome *bc_1_* complex (cytochrome *b*, cytochrome *c_1_*), as well as in cytochrome c loosely associated with the inner membrane of the mitochondrion. Cytochromes *c* family (cyt *c*) contains heme *c* covalently attached to the peptide backbone via one or two thioether bonds. Based on the size, number of heme groups and redox potential cytochromes *c* proteins are divided into four groups: class I-class IV. The low-spin soluble cyt *c* of mitochondria in which the heme-attachment site is located towards the N-terminus, and the sixth ligand located towards the C-terminus belongs to class I. Cytochrome *c* transfers electrons from complex III to complex IV, also known as cytochrome oxidase, which is the final enzyme of the electron transport system. The complex IV contains two cytochromes *a* and *a_3_* and two copper centers. Cytochrome *c*, which is reduced to cyt *c* Fe^2+^ (ferrous) by the electron from the complex III, passes an electron to the copper binuclear center, being oxidized back to cytochrome *c* (cyt *c* Fe^3+^ (ferric)). Using different excitations being in resonance with the absorption bands of the cytochromes one can spectrally enhance various cytochrome chromophores in the complex III, cytochrome *c* and complex IV by Resonance Raman enhancement scattering. Using different excitations, we are able to monitor contribution of cytochrome family in the electron transfer chain in the mitochondrial respiration originating from cytochrome *b*, *c_1_* in complex III, cytochrome *c*, and *a*, *a_3_* cytochromes in complex IV. The spectral features of cytochromes *a* and *a_3_* in complex IV can be observed at resonance conditions of 420 nm and 445 nm for the oxidized and reduced state, respectively as well as at pre-resonance conditions at other excitations. The spectral features of cytochrome *c* can be monitored at 532 nm due to absorbance band (Q band) centered at 530 nm [22,23,24]. The enhancement effect can be obtained by using different approach: different laser lines or by tunning the difference between two lasers with the targeted frequency.

Monitoring redox state of mitochondrial cytochromes has been demonstrated as a versatile clinical diagnostic tool with numerous successful reports on the detection of cancerous tissues in human patients [25,26,27,28].

It is evident that the real progress in improved cancer therapy and treatment depends on much better than today understanding biological mechanisms of mitochondrial dysfunction in cancer governed by balance between mechanisms for energy supply, macromolecular building blocks and maintenance of redox balance. To achieve this goal there is an urgent need to improve the conventional methods of molecular biology (immunohistochemistry, real-time PCR, immunoblotting, measurement of mitochondrial membrane potential, cell proliferation assay and caspase 3 activity assay) under hypoxic conditions in cancer diagnostics and develop imaging tools, because current imaging methods are often limited by inadequate sensitivity, specificity, spatial and spectral resolutions [29]. No technology has proven effectiveness for detecting invasive cancer. Existing clinical technologies-including state-of-the-art surgical microscopy, fluorescence-guided surgery and magnetic resonance imaging (MRI, used either pre- or intra-operatively)-cannot detect the full extent of cancer invasion [7]. The imaging methods that use radio-labelled ligands are not effective for detecting malignant gliomas. It has been reported that thirty five to forty percent of recurrent gliomas in human patients are not observed using positron emission tomography imaging techniques based on fluorodeoxyglucose uptake (e.g., FDG-PET), despite being detected by contrast MRI [4,30].

Our goal was to demonstrate possibility of monitoring redox state changes occurring in mitochondrial cytochromes in cancers. It is well known that cytochrome *c* is undoubtedly one of the most prominent molecules in the electron transport chain required to fuel life via ATP, but its role in molecular mechanisms associated with an aggressive phenotype of cancer remain largely unclear.

Therefore, this paper presents a truly unique landscape of cancer modern biochemistry by a non-invasive Raman approach to study redox status of cytochromes in brain and breast by means of Raman microspectroscopy, imaging at 355 nm, 532 nm, 633 nm, 785 nm excitations. In this paper we explore a hypothesis involving the role of reduction-oxidation pathways related to cytochrome *c* in cancer state. A thorough understanding of cytochrome role in brain and breast cancers with new methods will help establish Raman spectroscopy as a competitive clinical diagnosis tool for cancer diseases involving mitochondrial dysfunction.

## 2. Materials and Methods

### 2.1. Ethics Statement

All the conducted studies were approved by the local Bioethical Committee at the Polish Mother’s Memorial Hospital Research Institute in Lodz (53/216) and by the institutional Bioethical Committee at the Medical University of Lodz, Poland (RNN/323/17/KE/17/10/2017) and (RNN/247/19/KE).

Written consents from patients or from legal guardians of patients were obtained. All the experiments were carried out in accordance with Good Clinical Practice and with the ethical principles of the Declaration of Helsinki. Spectroscopic analysis did not affect the scope of surgery and course and type of undertaken hospital treatment.

### 2.2. Patients

For breast cancers the number of patients was 39, all patients were diagnosed with infiltrating ductal carcinoma and treated at the M. Copernicus Voivodeship Multi-Specialist Center for Oncology and Traumatology in Lodz. The overall human breast sample statistic was: n(G0) = 22, n(G1) = 3, n(G2) = 9, n(G3) = 5, where G0 describes healthy tissue, G1, G2, G3 the cancer tissues with increasing malignancy grade. 

The total number of patients diagnosed with brain tumors was 43. Among patients with brain tumors 11 were diagnosed with medulloblastoma, one with embryonic tumor PNS, three with ependynoma anaplastic, four with ependymoma, two with astrocytoma fibrous, one with astrocytoma, one with ganglioma, eight with astrocytoma pilocytic, one with subependymoma, two with hemangioblastoma, four with craniopharyngioma, one with dysembryoplastic neuroepithelial tumor, one with papillary glioneuronal tumor, one with tumor metastasis, one with anaplastic oligodendroglioma and one with meningioma. 

All patients for brain cancers were treated at the Polish Mother’s Memorial Hospital Research Institute in Lodz and at the Department of Neurosurgery, Spine and Peripheral Nerve Surgery of University Hospital WAM-CSW in Lodz. The overall human brain samples statistic was: n(G0) = 1, n(G1) = 17, n(G2) = 9, n(G3) = 4, n(G4) = 13, where G0 describes healthy tissue, G1, G2, G3, G4 the cancer tissues with increasing malignancy grade 

### 2.3. Materials

Cytochrome *c* from the equine heart (C7752, Sigma-Aldrich, Poznań, Poland) was used without additional purification.

### 2.4. Cell Culture and Preparation for Raman Spectroscopy and Fluorescence Imaging

The studies were performed on a human adenocarcinoma cell line AU656 (ATCC CRL-2351) purchased from American Type Culture Collection (ATCC, Kiełpin, Poland). The AU656 cells were maintained in RPMI-1640 medium (ATCC 30-2001) supplemented with 10% fetal bovine serum (ATCC 30-2020) without antibiotics in a humidified incubator at 37 °C and 5% CO_2_ atmosphere. Cells were seeded on CaF_2_ window (Crystran Ltd., Poole, UK; CaF_2_ Raman grade optically polished window 25 mm dia x 1 mm thick, no. CAFP25-1R, Poole, UK) in 35 mm Petri dish at a density of 5 × 10^4^ cells per Petri dish the day before the examination.

Before Raman examination, the growing medium was removed and the cells were fixed with 10% formalin for 10 min and kept in phosphate-buffered saline (PBS, no. 10010023, Gibco, Warszawa, Poland) during the experiment. After the Raman imaging measurements, the cells were exposed to Hoechst 33342 (25 μL at 1 μg/mL per mL of PBS) and Oil Red O (10 μL of 0.5 mM Oil Red dissolved in 60% isopropanol/dH_2_O per each mL of PBS) by incubation for 15 min. The cells were then washed with PBS, followed by the addition of fresh PBS for fluorescence imaging on a WITec Alpha 300RSA+ microscope (WITec, Ulm, Germany).

### 2.5. Tissues Samples Collection and Preparation for Raman Spectroscopy

Tissue samples were collected during routine surgery. The non-fixed samples were used to prepare 16 micrometers sections placed on CaF_2_ substrate for Raman analysis (Crystran Ltd, CaF_2_ Raman grade optically polished window 25 mm dia × 1 mm thick, no. CAFP25-1R). In parallel typical histopathological analysis by professional pathologists from the Polish Mother’s Memorial Hospital Research Institute in Lodz for brain tissues samples or from Medical University of Lodz, Department of Pathology, Chair of Oncology for breast tissues samples was performed. The types and grades of tumors according to the criteria of the Current WHO Classification were diagnosed by professional and certified histopatologists.

### 2.6. Raman Human Cell and Tissue Spectroscopic Measurements In- and Ex-Vivo

A WITec Alpha 300 RSA+ confocal microscope was used to record Raman spectra and imaging. The configuration of experimental set up was as follows: the diameter of fiber: 50 μm for 355 and 532 nm and 100 μm for 785 nm, an Acton-SP-2300i monochromator (Acton, MA, USA) and a DU970-UVB-353 CCD camera for 355 and 532 nm (Andor Newton, Belfast, Northern Ireland) and an Ultra High Throughput Spectrometer (UHTS 300, WITec, Ulm, Germany) and an Andor Newton iDU401A-BR-DD-352 CCD camera for 785 nm. Excitation lines were focused on the sample through a 40x dry objective (Nikon, Warszawa, Poland) objective type CFI Plan Fluor C ELWD DIC-M, numerical aperture (NA) of 0.60 and a 3.6–2.8 mm working distance). The average laser excitation power was 1 mW for 355 nm, 10 mW for 532, 10 mW for 633 nm, and 80 mW for 785 nm, with an integration time of 0.5 s, 0.5 s/1.0 s, 2.0 s and 0.5 s respectively. An edge filters were used to remove the Rayleigh scattered light. A piezoelectric table was used to record Raman images. No pre-treatment of the samples was necessary before Raman measurements. The cosmic rays were removed from each Raman spectrum (model: filter size: 2, dynamic factor: 10) and the smoothing procedure: Savitzky–Golay method was also implemented (model: order: 4, derivative: 0). Data acquisition and processing were performed using the WITec Project Plus software.

### 2.7. Statistical Analysis

All results regarding the analysis of the intensity of the Raman spectra as a function of breast cancer or brain tumor grades are presented as the mean ± SD, where *p* < 0.05; SD - standard deviation, p – probability value. Raman bands intensity were taken from normalized by vector norm spectra. The Raman spectra were obtained from 39 patients (breast) and 44 (brain). For each patient thousand spectra from different sites of the sample were obtained from cluster analysis. For breast we used typically 6400 Raman spectra for averaging, and 4900 Raman spectra for brain tissue. 

To show the perfect match between Raman spectra of human tissue samples and Raman spectrum of cytochrome *c* the correlation analysis was performed (Pearson correlation coefficient was equal 0.99951 at the confidence level 0.95 with the *p*-value of 0.00001).

### 2.8. Cluster Analysis

Spectroscopic data were analysed using the cluster analysis method. Briefly cluster analysis is a form of exploratory data analysis in which observations are divided into different groups that have some common characteristics – vibrational features in our case. cluster analysis constructs groups (or classes or clusters) based on the principle that: within a group the observations must be as similar as possible, while observations belonging to different groups must be as different.

The partition of n observations (x) into k (k ≤ n) clusters S should be done to minimize the variance (Var) according to the formula:
$$\arg min_{S}\overset{k}{\underset{i=1}{\sum }}\underset{x\in Si}{\sum }∥x\;\mu_{i}\;{∥}^{2}=\arg min_{S}\overset{k}{\underset{i=1}{\sum }}\left|S_{i}\right|VarS_{i}$$ where $\mu_{i}$ is the mean of points $S_{i}$.

Raman maps presented in the manuscript were constructed based on principles of cluster analysis described above. Number of clusters was six (the minimum number of clusters characterized by different average Raman spectra, which describe the variety of the inhomogeneous biological sample). The colors of the clusters correspond to the colors of the average Raman spectra of lipids (blue), proteins (red), cytochrome (green), mitochondria (magenta), lipids and proteins (dark green and gray).

## 3. Results

To properly address redox state changes of mitochondrial cytochromes in brain and breast cancers by Raman spectroscopy and imaging, we systematically investigated how the Raman method responds to redox upregulations in cancers. The *ex vivo* human tissue experiments will extend our knowledge on the influence of cytochrome family on cancer development.

First, we compared the Raman spectra of the human brain and breast cancer tissues using different laser excitation wavelengths. This approach might generate Raman resonance enhancement for some tissue components that cannot be visible for non-resonance conditions and provide selective spectral isolation of components crucial for understanding mechanisms of mitochondrial dysfunctions associated with cancers [31]. Thus, using different excitations one can learn about different aspects of cancer development.

Figure 1 shows the average Raman spectra of human brain tissue of medulloblastoma (grade of malignancy WHO G4) at different excitations (number of patients n = 6) of surgically resected ex vivo tumor human brain tissue at the various excitation 532 nm, 355 nm, 785 nm, for the same area of the samples.

Figure 2 shows the average Raman spectra of the ex vivo human breast cancer tissue surgically resected specimens, ductal cancer, grade of malignancy WHO G3 at different excitations 532 nm, 633 nm, 785 nm, (number of patients n = 5), and Raman spectrum of the pure cytochrome c at 532 nm excitation.

Figure 1 and Figure 2 reveal that the Raman spectra of cancer tissue corresponding to different excitations are significantly different indicating that the resonance enhancement of Raman scattering occurs in the tissue. Different vibrations are enhanced at different excitation wavelengths.

The excitation at 532 nm enhances two types of components of the tissue: carotenoids (1520 cm^−1^ and 1158 cm^−1^) and cytochromes *c* and *b* (750, 1126, 1248, 1310, 1337, 1352, 1363, 1584, and 1632 cm^−1^) [23,24]. As the enhancement of carotenoids was discussed in our laboratory in many previous papers [32,33,34,35,36,37], here we will concentrate on cytochrome family. Figure 2B shows the spectrum of isolated cytochrome *c*. Using 532 nm laser excitation one can monitor spectral features of complex III and cytochrome *c* due to Q bands at 500-550 nm related to intra-porphyrin transitions of the heme group in cytochrome *c* [38,39].

Excitation at 633 nm provides information about cytochromes *a* and *a_3_* (1744 cm^−1^ and 1396 cm^−1^, both in cyt oxidized and reduced cytochrome oxidase; 1584 cm^−1^, heme *a + a_3_* oxidized form) [22].

The excitation at 785 nm is far from resonances of cytochromes and represents other compounds of the tissue, which are not clearly identified.

First, let us concentrate on the contribution of cytochrome *c* using 532 nm excitation. Figure 3 shows the average (n = 44), Raman spectra of ex vivo human normal (grade of malignancy G0) and tumor brain tissue (G1, G2, G3, G4) and breast tissue (n = 39) (G0, G1, G2, G3) surgically resected specimens at 532 nm.

Since 532 nm excitation causes resonance Raman scattering from both *c* and *b*-type cytochromes due to electronic Q band absorption cytochromal Raman peaks may originate from both cytochrome types.

Comparison of the Raman spectra of the brain tissue in Figure 1 and the Raman spectra of the breast tissue in Figure 2A with purified cytochrome *c* in Figure 2B shows that the Raman enhancement upon 532 nm excitation corresponds perfectly to vibrations of cytochrome c [23,24]. To show the perfect match between Raman spectra of human tissue samples and Raman spectrum of cytochrome *c* the correlation analysis was performed (Pearson correlation coefficient was equal 0.99951 at the confidence level 0.95 with the *p*-value of 0.00001). It indicates that cytochrome *c* can be used for pathology assessment for ex vivo tissues.

To check if vibrations of cytochrome *c* can be used also for pathology assessment in living cells we analysed Raman spectra of brain and breast cancer cells lines at in vitro incubation. Figure 4 shows confocal Raman spectroscopy analysis of the human adenocarcinoma cell line (invasive ductal cancer (AU565)), receptor expression: epidermal growth factor (EGF), Oncogene: her2/neu+ (overexpressed); her3+; her4 +; p53+), microscopy image, Raman image from the cluster analysis: nucleus (red), endoplasmic reticulum (blue), lipid droplets (orange) cytoplasm (green), mitochondria (magenta), cell border (light grey), area out of the cell (dark grey), at the 532 nm wavelength excitation.

We compared Raman spectra of single cells at in vitro incubation and Raman spectrum of cytochrome *c*. To show the perfect match between Raman spectra of human cells and Raman spectrum of cytochrome *c* the correlation analysis was performed (Pearson correlation coefficient was equal 0.99941 at the confidence level 0.95 with the *p*-value of 0.00002). It indicates that cytochrome *c* can be used for pathology assessment for living cells. 

Literature assignments [20,39,40,41] show that some cytochromal Raman peaks are common to *c*, *c_1_* and *b* cytochromes. Thus, the major peaks at 750 and 1126 cm^−1^ are present in both types of cytochromes, whereas the peaks at 1310 cm^−1^ and 1398 cm^−1^ correspond to *c*-type cytochromes and the peaks at 1300 and 1337 cm^−1^ - to *b*-type cytochromes. Thus, the peak at 1337 cm^−1^ can be useful to distinguish between cytochrome *c* and *b*, as the vibration at 1337 cm^−1^ represents a unique peak of the reduced cyt *b* (ferrous (Fe^2+^) cytochrome). Therefore, the peaks at 750 and 1126 cm^−1^ observed in Raman spectra of the brain and breast tissues in Figure 3 represent both *c*, *c_1_* and *b*-types of cytochromes. However, relative contributions of cyt *c* and cyt *b* to the overall Raman band differ in biological systems. It was reported [20,39,40,41] that under 530.9 nm laser excitation the Raman peak at 750 cm^−1^ was mainly determined by *c*-type cytochromes, whereas peak at 1126 cm^−1^ by *b*-type cytochromes. Hence, the ratio of intensities 750/1126 can be used to estimate the relative amount of reduced cytochromes *c*, *c_1_* vs. reduced cytochromes *b*.

The vibrations of cytochrome c that are resonance Raman enhanced in the brain and breast tissues are demonstrated (with green arrows) in Figure 1, Figure 2 and Figure 3. We observe four intensive peaks: 750 (symmetric vibrations of pyrrole rings), 1126 (vibrations of C_b_-CH_3_ side radicals), 1310 (vibrations of all heme bonds), 1363 (mode ν_4_) and 1584 cm^−1^ (v_19_ mode, vibrations of methine bridges (CαCμ, CαCμH bonds) and the CαCβ bond). There are also a number of other peaks with lower intensities 1248, 1352, 1632 cm^−1^ (methine bridges (bonds CαCμ, CαCμH). The Raman bands of the reduced form have higher intensities [24]. Symmetric vibrational modes of the porphyrin ligand in cytochrome c are resonance Raman enhanced to a greater degree using excitation wavelengths within the Soret absorption peaks at 408 nm (ferric, Fe^3+^), 416 nm (ferrous, Fe^2+^) states, whereas asymmetric modes are enhanced to a greater degree using excitation wavelengths within the Q absorption peak at 500-550 nm [42]. Detailed vibrational assignment of cytochrome *c* can be found in ref [24] Q-resonant Raman spectra contain unusually strong depolarized bands. In fact, the B1g pyrrole breathing mode v_15_ (750 cm^−1^) gives rise to one of the strongest bands (Figure 2B). The bands of cytochrome c at 750, 1126, 1248, 1310, 1363 cm^−1^ are depolarized and represent the reduced form. Anomalously polarized bands appear in the Q-resonant spectra. Especially striking is the v_19_ mode [24] (1584 cm^−1^), which produces one of the most prominent bands in the perpendicularly polarized spectrum. The band at 1584 cm^−1^ represents the reduced form of cytochrome *c* and it is not observed in the oxidized form. Some of the peaks of the oxidized form of cytochrome *c* (around 750, 1130, 1172, 1314, 1374, 1570–1573 and 1634 cm^−1^) have the same positions as the reduced form [43], but their intensities are significantly lower except the band 1634 cm^−1^ corresponding to the ferric cytochrome *c* as presented in Figure 5.

Figure 5 shows the electronic absorption spectra and the Raman intensities of the reduced form of cyt *c* Fe^2+^ and the oxidized form cyt *c* Fe^3+^ and the electronic absorption spectra. Our results in Figure 5 show that the Raman intensities of the reduced form of cytochrome *c* (cyt *c* Fe^2+^) are significantly higher than those of the oxidized form (cyt *c* Fe^3+^) and they support earlier results presented in the literature [19,43,44,45].

We used 1584 cm^−1^ vibrational mode (ν_19_) as a marker band of ferrous cyt *c* in brain and breast cancer tissues (Figure 1, Figure 2 and Figure 3). Although there are several overlapping bands in that region: ν_19_ of ferric heme c (1582 cm^−1^), ν_19_ of ferrous heme c (1582 cm^−1^), ν_2_ of ferric heme *c* (1585 cm^−1^), ν_19_ of ferrous heme cyt *b* (1586 cm^−1^) and ν_2_ of ferrous heme *b* (1583 cm^−1^) we can eliminate from our discussion all ferric modes due to the fact that the resonance Raman intensities of the ferric modes are very weak in comparison to the ferrous bands except the band 1634 cm^−1^ corresponding to the ferric cytochrome *c*. Thus, the bands at 1584 cm^−1^ (reduced cyt *c*) and at 1634 cm^−1^ (oxidized cyt *c*) of cytochrome *c* can be used as a very important parameter controlling the level of reduction in cancer tissues.

The Raman spectra in Figure 1, Figure 2 and Figure 3 do not provide information about the distribution of cytochrome *c* in the cancer tissue. To learn about the distribution of cytochrome *c* we used Raman polarized imaging. Polarized Raman spectroscopy [46] provides vital information about a molecular structure, orientation in highly ordered systems and conformational preferences of a specific crystalline structure.

Figure 6 shows the Raman spectra and images for the human breast tissue of invasive ductal carcinoma (IDC) at different experimental geometries for Raman scattering: without a polarization analyzer, and at parallel I_II_ and perpendicular polarizations 
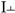
 of the incident and Raman scattered beams.

One can see from Figure 6 that the vibrational mode at 1584 cm^−1^ shows anomalously polarized bands appearing in the Q-resonant spectra at 532 nm, which produces one of the most prominent bands in the perpendicularly polarized spectrum. The anomalously polarized band at 1584 cm^−1^ appearing in the Q-resonant Raman spectra is characteristic of cytochrome *c* [24]. This result additionally supports our previous findings from Figure 1 and Figure 2 that the Q-resonant Raman band at 1584 cm^−1^ in the breast and brain tissues represents cytochrome *c*.

Figure 6 shows microscopy image, Raman image, and the average Raman spectra of lipids (blue), proteins (red), cytochrome (green), mitochondrion (magenta) at 532 nm at different experimental geometries for Raman scattering: without polarization analyzer, at parallel and perpendicular polarizations of the incident and Raman scattered beams for infiltrating ductal cancer (malignancy grade G2).

One can see from Figure 6E that the band at around 1584 cm^−1^ is one of the most prominent bands in the perpendicularly polarized spectrum. Since Raman scattering at 1584 cm^−1^ is not observed in the oxidized form of cytochrome *c* the band at 1584 cm^−1^ originates from the reduced form of cytochromes *c* [19].

Our results demonstrate that the Raman intensity of 1584 cm^−1^ peak is the most sensitive vibration of redox status in the cell and is related to the amount of the reduced cytochromes *c*. Therefore, we used 1584 cm^−1^ to study the redox state of mitochondrial cytochrome *c* in brain and breast ex vivo human tissues.

To check whether the redox state of cytochrome *c* is related to cancer aggressiveness we used the Raman redox state biomarker represented by the Raman intensity of 1584 cm^−1^ peak.

Creating Raman images as presented in Figure 6B one can analyze distribution of proteins (red color), lipids profiles (blue color), and cytochrome (green color) and learn about the biochemical composition from the corresponding Raman spectra presented in Figure 6C–E.

In the view of the results presented so far we can state that Raman spectroscopy can be used in probing biochemical changes in normal and cancer cells, and the role of cytochromes in these specific cancers, which is not previously reported.

To check whether cytochromes are upregulated in brain and breast cancers we studied the Raman signals corresponding to cytochromes as a function of cancer malignancy described by using grades. The grade is based on how much the cancer cells look like normal cells and measure cell anaplasia (reversion of differentiation). The grade is used to help predict outcome (prognosis) and to help figure out what treatments might work best. Grading systems are also different for brain and breast types of cancer. The following pattern with grades being increasingly malignant over a range of 1 to 4 is used for brain tumors and 1 to 3 for breast tumors [47]. If no specific system is used, the following general grades are most commonly used for brain:GX Grade cannot be assessedG1 Well differentiated (Low grade)G2 Moderately differentiated (Intermediate grade)G3 Poorly differentiated (High grade)G4 Undifferentiated (High grade)
and for breast:Grade 1 or well differentiated (score 3, 4, or 5). The cells are slower-growing, and look more like normal breast tissue.Grade 2 or moderately differentiated (score 6, 7). The cells are growing at a speed of and look like cells somewhere between grades 1 and 3.Grade 3 or poorly differentiated (score 8, 9). The cancer cells look very different from normal cells and will probably grow and spread faster.

Figure 7A shows the intensity of the 750, 1126, 1337 and 1584 cm^−1^ Raman peak of cytochrome *c* as a function of grade. Figure 7B shows the data for 1584 cm^−1^ band of reduced cytochrome *c* for breast normal (G0) and cancer (invasive ductal cancer) human tissue (G1, G2, G3). Based on the average values (number of patients n = 39) obtained for the Raman biomarker of cytochrome *c* I_1584_ we obtained a plot as a function of cancer grade malignancy.

In the view of the results presented in Figure 7 it is evident that the Raman biomarker I_1584_ measuring contribution of cytochrome *c* in the human breast tissues correlates with breast cancer aggressiveness. The intensity of the 1584 cm^−1^ Raman signal corresponding to the amount of reduced cytochrome c increases with increasing cancer aggressiveness. It indicates that cytochrome *c* plays a crucial role in the development and progression of cancer. The parabolic dependence of the Raman biomarker I_1584_ of the reduced cytochrome *c* in Figure 7 vs cancer malignancy shows that the optimal concentration of cytochrome *c* that are needed to maintain cellular homeostasis corresponds to the normalized intensity of 0.006 ± 0.003 for the breast tissues. The concentrations of the reduced cytochrome *c* at this level modulate protective, signaling-response pathways, resulting in positive effects on life-history traits. The reduced cytochrome *c* level above the value of 0.06 ± 0.02 triggers a toxic runaway process and aggressive breast cancer development.

The plot provides an important cell-physiologic response, normally the reduced cytochrome *c* operates at low, basal level in normal cells, but it is strongly induced to very high levels in pathological cancer states.

It is interesting to check if the other characteristic bands of cytochrome *c* and cytochrome *b* correlate with breast cancer grade malignancy G0-G3.

Figure 7B,C shows the Raman signals of other characteristic vibrations of cytochromes *c* and *b* as a function of breast cancer grade malignancy G0-G3 for the human tissues. One can see that the bands of mixed vibrations of cytochromes *c* and *b* (750 and 1126 cm^−1^) increases with aggressiveness. The peaks at 1337 cm^−1^ corresponding to cytochromes *b* does not practically change with breast cancer aggressiveness.

It was reported [20,39,40,41] that under 530.9 nm laser excitation the Raman peak at 750 cm^−1^ was mainly determined by *c*-type cytochromes, whereas peak at 1126 cm^−1^ by *b*-type cytochromes.

We analyzed the Raman intensity ratio of the peaks at 750 cm^−1^ and 1126 cm^−1^ to evaluated the relative contribution of cytochrome *c* and *b* (see Figure 7B,C). Figure 8 shows the Raman intensity ratio of the peaks at 750 cm^−1^ and 1126 cm^−1^ in human tissue as a function of breast cancer grade malignancy G0-G3 at excitation 532 nm.

One can see from Figure 8 that the relative intensity ratio I_750_/I_1126_ increases with breast cancer aggressiveness. It indicates that the relative contribution of cytochrome *c* is higher than cytochrome *b* when the grade of malignancy increases.

In the view of the results obtained for breast cancer we want to check if the same Raman biomarkers can be used to monitor cytochromes in brain tumors.

Figure 9 shows microscopy image, Raman image, and the average Raman spectra of lipids (blue), proteins (red), cytochrome (green), mitochondrion (magenta) at 532 nm for Anaplastic oligodendroglioma (WHO classification as malignancy grade G3) [48].

Figure 9C,F show that the intensity of the 1584 cm^−1^ Raman signal corresponding to the concentration of reduced cytochrome *c* increases with increasing brain tumor aggressiveness. Indeed, one can see that the Raman signal at 1584 cm^−1^ is significantly stronger for G3 (Figure 9C) than for G2 (Figure 9F).

To check whether cytochromes *c* is upregulated in brain tumors we studied the Raman signals at 1584 cm^−1^ corresponding to concertation of cytochrome *c* as a function of brain tumor malignancy by using WHO grade.

Figure 10 shows the Raman intensities of characteristic vibrations in brain tissue corresponding to concentration of reduced cytochrome *c*. One can see that for the brain tumors the Raman intensity of all characteristic vibrations of cytochrome *c* increases with increasing cancer aggressiveness up to G3 and then slightly decreases for G4. The amount of cytochrome *b* is practically the same for all grades of brain tumor.

## 4. Conclusions

We showed that Raman imaging provides additional insight into the biology of astrocytomas and breast ductal invasive cancer, which can be used for noninvasive grading, differential diagnosis, delineation of tumor extent, planning of surgery, and radiotherapy and post-treatment monitoring. We used Raman spectroscopy and Raman imaging to monitor changes in the redox state of the mitochondrial cytochromes in ex vivo human brain and breast tissues surgically resected by means of Raman microspectroscopy at 532 nm, 633 nm, 785 nm.

The results presented in the paper suggest that the redox-sensitive peak observed at 1584 cm^−1^ with excitation at 532 nm is specifically linked to cytochrome *c*. and can be considered to be a “redox state marker” of the ferric low spin heme in cyt *c*, assigned to the v_19_ mode, vibrations of methine bridges (CαCμ, CαCμH bonds) and the CαCβ bond.

Our results show that human breast cancer and brain tumor demonstrate a redox imbalance compared to normal tissues. We found that concentration of cytochrome *c* is upregulated in brain and breast cancers. The intensities of the 1584 cm^−1^ Raman signal corresponding to the amount of the reduced cytochrome *c* increases with increasing cancer aggressiveness. It indicates that cytochrome *c* plays a crucial role in the development and progression of cancer. We found the dependence of the Raman biomarker I_1584_ of the reduced cytochrome *c* vs grade of cancer malignancy which shows that the optimal concentration of cytochrome *c* that are needed to maintain cellular homeostasis corresponds to the normalized Raman intensity of 0.006 ± 0.003 for the breast tissue and 0.074 ± 0.005 for the brain tissue. The concentrations of the reduced cytochrome *c* at this level modulate protective, signaling-response pathways, resulting in positive effects on life-history traits. The reduced cytochrome *c* level above these values triggers a toxic runaway process and aggressive cancer development.

The relation between the Raman signal intensity at 1584 cm^−1^ of the reduced cytochrome *c* vs cancer grade provides an important cell-physiologic response demonstrating that the reduced cytochrome *c* operates at low, basal level in normal cells, but it is strongly induced to very high levels in pathological cancer states. We found that also the Raman intensity of the mixed vibrations of cytochromes *c* and *b* (750 and 1126 cm^−1^) increase with aggressiveness. The Raman peaks at 1337 cm^−1^ corresponding to cytochromes *b* does not practically change with breast and brain cancer aggressiveness. We analyzed the Raman intensity ratio of the peaks at 750 cm^−1^ and 1126 cm^−1^ to evaluate the relative contribution of cytochrome *c* and *b* in human breast tissue as a function of breast cancer grade malignancy G0-G3 at excitation 532 nm. We found that the relative contribution of cytochrome *c* is higher than cytochrome *b* when the grade of malignancy increases.

## Figures and Tables

**Figure 1 cancers-13-00960-f001:**
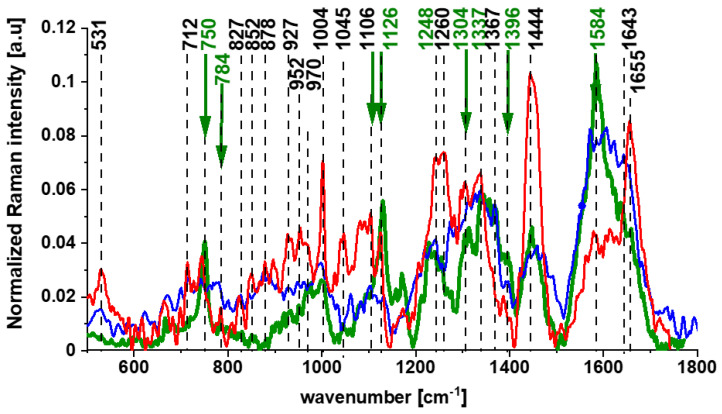
The average Raman spectra for the human brain tissue of medulloblastoma (grade of malignancy WHO G4) at different excitations (number of patients n = 6, for each patient thousands of Raman spectra obtained from cluster analysis) of the ex vivo tumor human brain tissue of medulloblastoma at the excitations 355 nm ▬ (blue), 532 nm ▬ (green) and 785 nm ▬ (red) for the same area of the samples.

**Figure 2 cancers-13-00960-f002:**
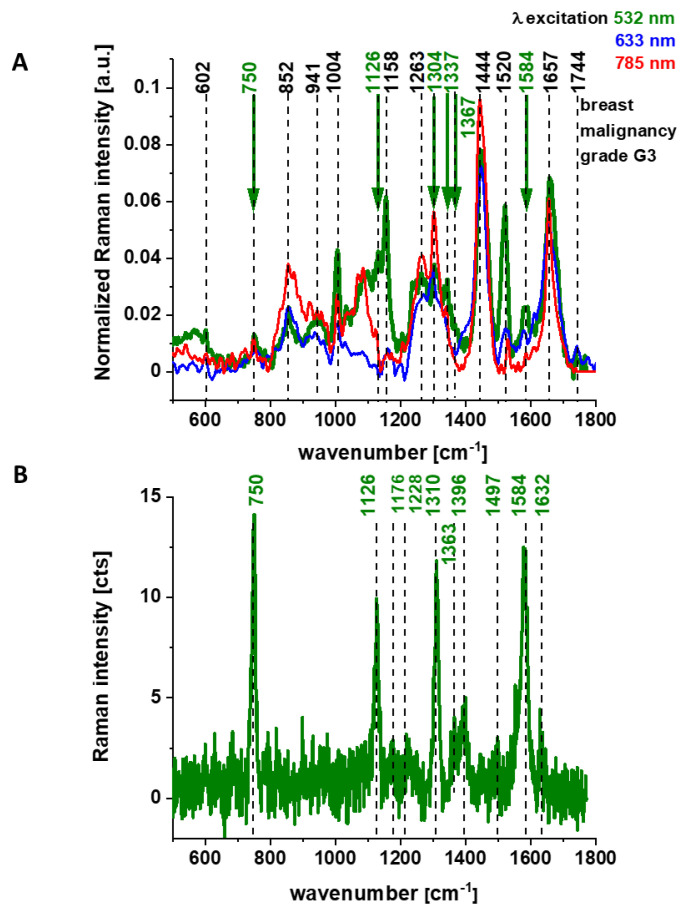
The average Raman spectra of the ex vivo human breast cancer tissue surgically resected specimens, ductal cancer, grade of malignancy WHO G3 at the excitations 633 nm ▬ (blue), 532 nm ▬ (green) and 785 nm ▬ (red) (number of patients n = 5, for each patient thousands of Raman spectra obtained from cluster analysis) (**A**), Raman spectrum of the pure cytochrome *c* at 532 nm excitation (**B**).

**Figure 3 cancers-13-00960-f003:**
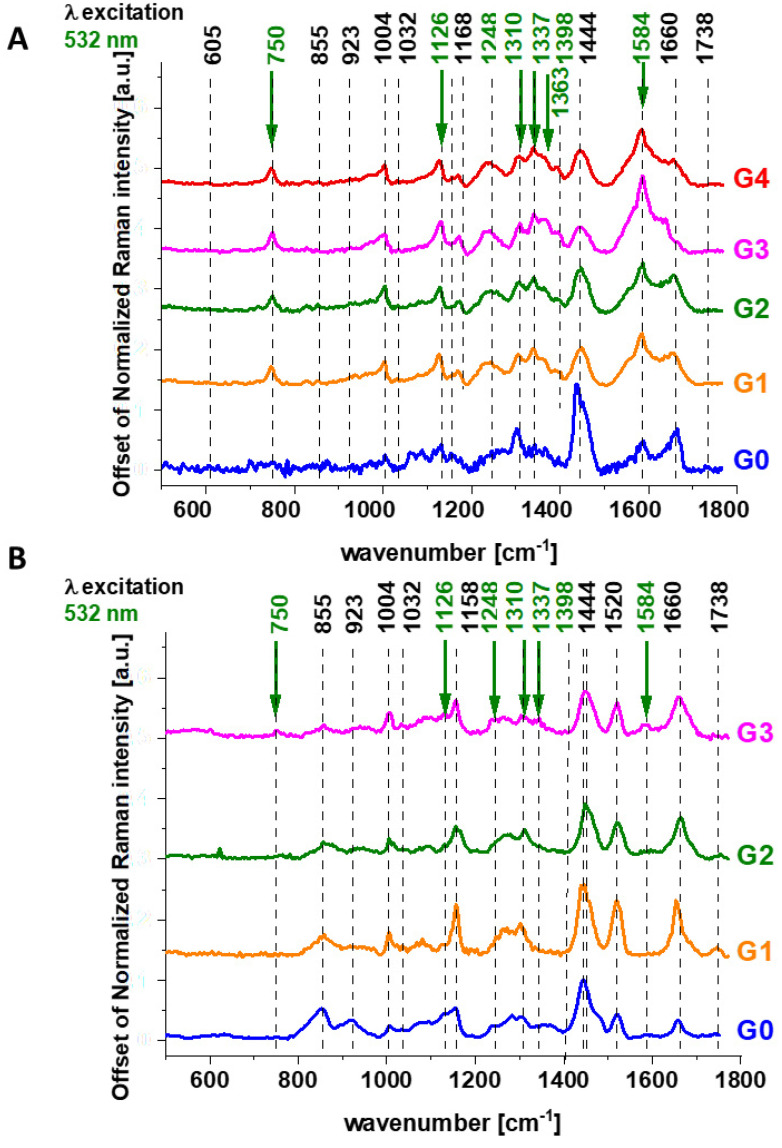
The average Raman spectra of ex vivo human normal (grade of malignancy G0) and tumor G1. G2, G3, G4 brain tissue (n = 44) (**A**) and ex vivo human normal (grade of malignancy G0) and G1, G2, G3 breast tissue (n = 39) (**B**), surgically resected specimens at 532 nm.

**Figure 4 cancers-13-00960-f004:**
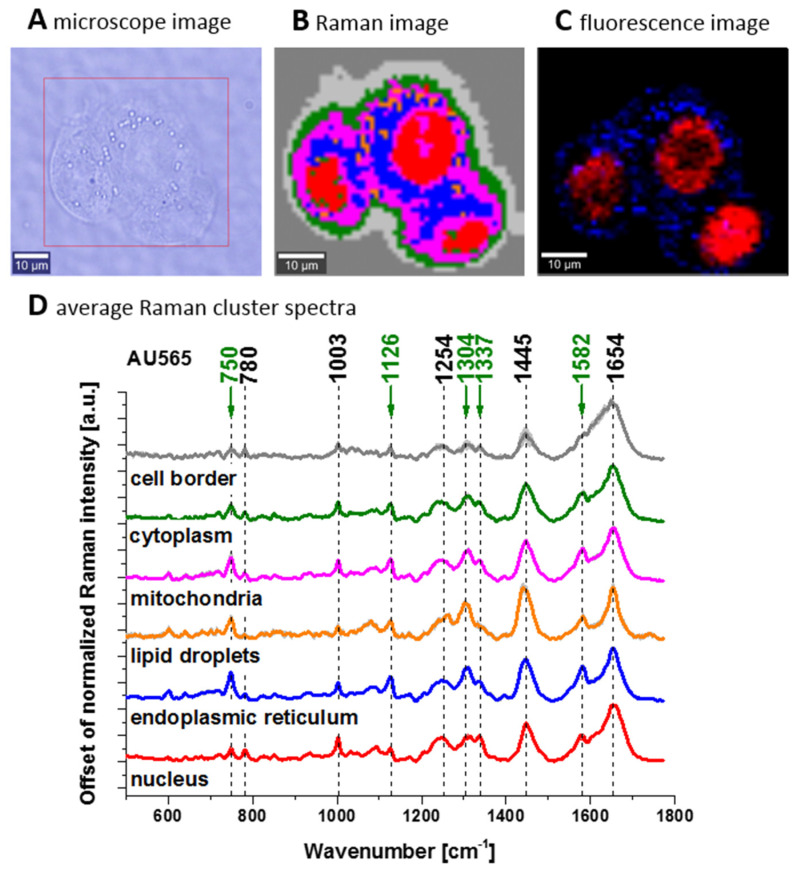
Confocal Raman spectroscopy analysis of the human adenocarcinoma cell line (invasive ductal cancer (AU565)) at the 532 nm wavelength excitation. (**A**) Microscopy image, (**B**) Raman image from the cluster analysis (nucleus (red), endoplasmic reticulum (blue), lipid droplets (orange) cytoplasm (green), mitochondria (magenta), cell border (light grey), area out of the cell (dark grey), image size: 55 μm × 50 μm, resolution of 1 μm, laser excitation 532 nm, power 10 mW, integration time 0.3 s), (**C**) fluorescence image of lipids (blue, Oil Red O staining) and nucleus (red, Hoechst 33342 staining). (**D**) Average Raman cluster spectra for the number of cells, n = 20 (8639 spectra were recorded (n(nucleus) = 2142, n(endoplasmic reticulum) = 1530, n(lipid droplets) = 121, n(cytoplasm) = 1464, n(mitochondria) = 1689, n(cell border) = 1693).

**Figure 5 cancers-13-00960-f005:**
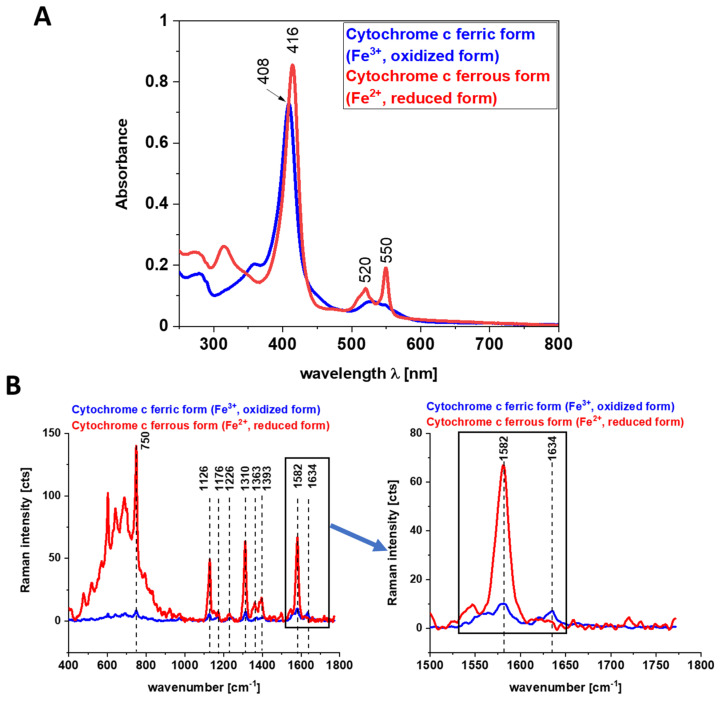
Electronic absorption spectra (**A**) and Raman spectra (**B**) of cytochrome *c* in ferric (oxidized, Fe^3+^) and ferrous (reduced, Fe^2+^) states in phosphate buffer pH = 7.3, cuvette optical path 1 cm. Ferrous cytochrome *c* was prepared by adding 10-fold excess NaBH_4_ (as a reductor).

**Figure 6 cancers-13-00960-f006:**
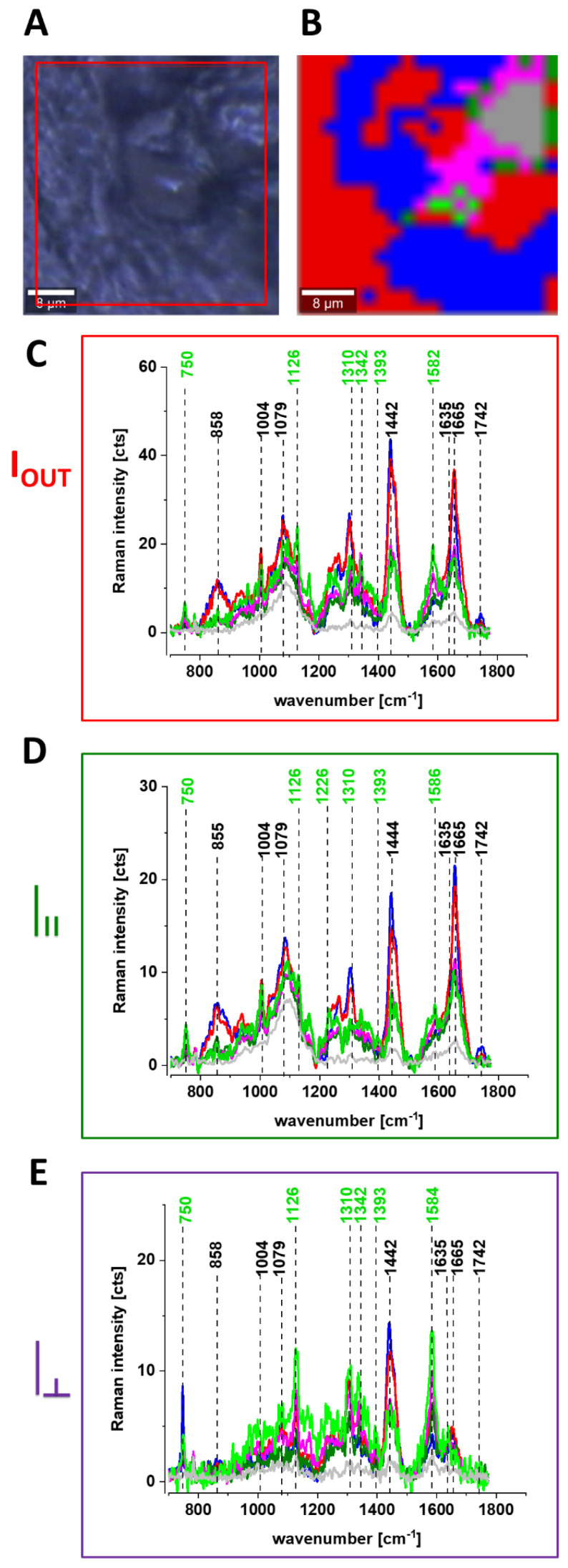
Microscopy image (**A**), Raman image (40 × 40 μm, resolution 0.5 μm, integration time 1.0 s) (**B**), the average Raman spectra of lipids ▬ blue, proteins ▬ red, cytochrome ▬ green, mitochondrion ▬ magenta at 532 nm at different experimental geometries for Raman scattering: without polarization analyzer (**C**), at parallel (**D**) and perpendicular (**E**) polarizations of the incident and Raman scattered beams for P 129 Infiltrating ductal carcinoma G2.

**Figure 7 cancers-13-00960-f007:**
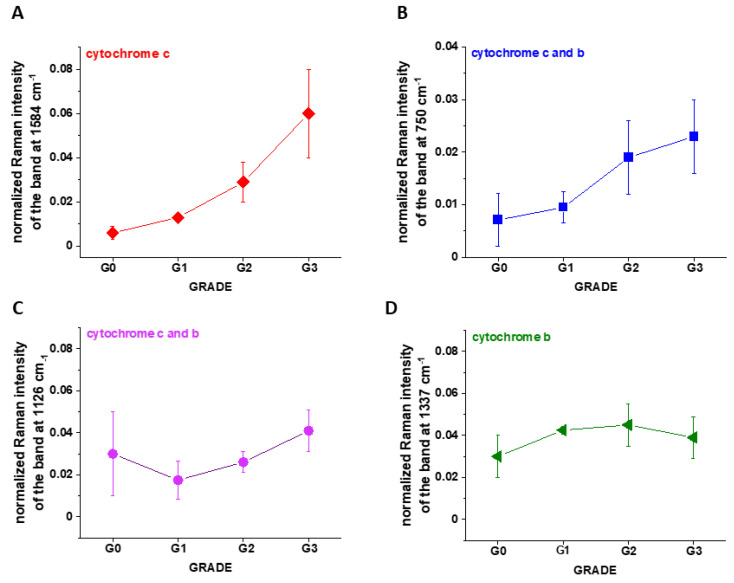
The Raman intensities of cytochrome *c* and cytochrome *b* in human breast tissue for n = 39: I_1584_ (**A**), I_750_ (**B**), I_1126_ (**C**), I_1337_ (**D**) as a function of breast cancer grade malignancy G0-G3 at excitation 532 nm (A). The results are presented as the mean ± SD. Raman bands intensity were taken from normalized by vector norm spectra.

**Figure 8 cancers-13-00960-f008:**
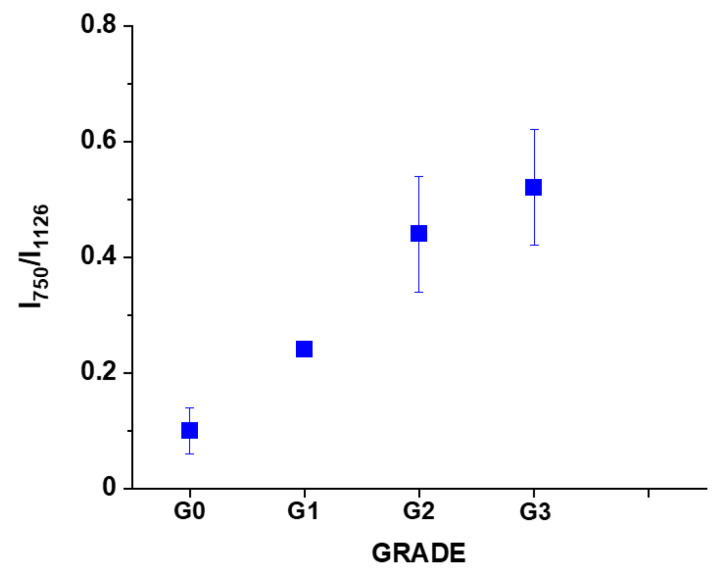
The Raman intensity ratio of the peaks at 750 cm^−1^ and 1126 cm^−1^ I_750_/I_1126_ in human breast tissue as a function of breast cancer grade malignancy G0-G3 at excitation 532 nm. The results are presented as the mean ± SD. Raman bands intensity were taken from normalized by vector norm spectra.

**Figure 9 cancers-13-00960-f009:**
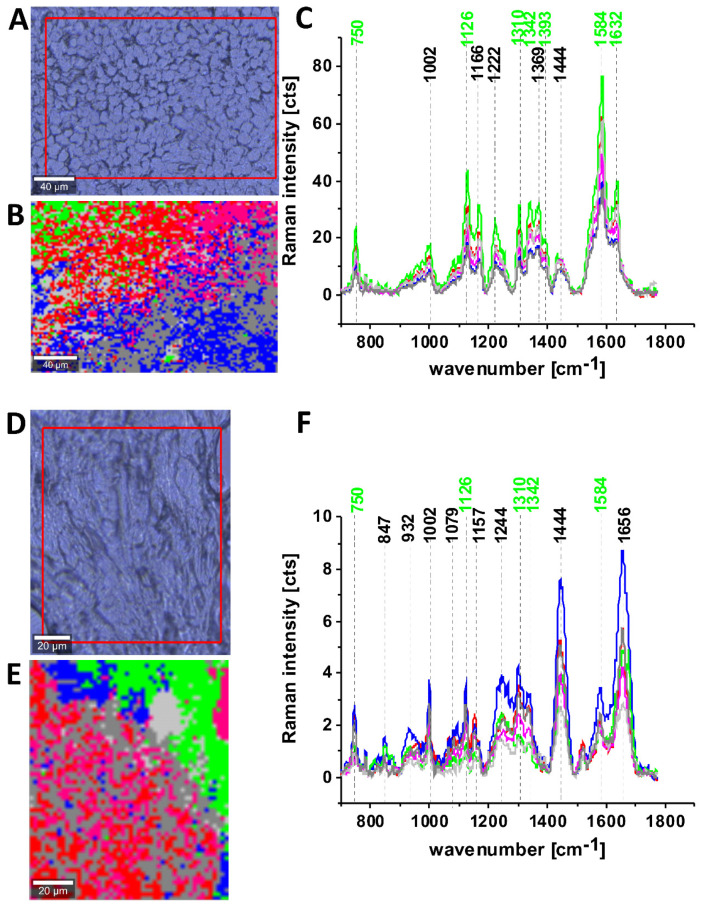
Microscopy image (**A**), Raman image (230 × 160 µm, resolution 0.5 µm, integration time 1 s) (**B**), the average Raman spectra of lipids (blue color), proteins (red color), cytochrome (green color), mitochondria – magenta at excitation 532 nm, Anaplastic oligodendroglioma (G3) (**C**) Microscopy image (**D**), Raman image (100 × 120 µm, resolution 0.5 µm, integration time 1 s) (**E**), the average Raman spectra of lipids (blue color), proteins (red color), cytochrome (green color), mitochondria (magenta color) at excitation 532 nm, meningioma (G2) (**F**).

**Figure 10 cancers-13-00960-f010:**
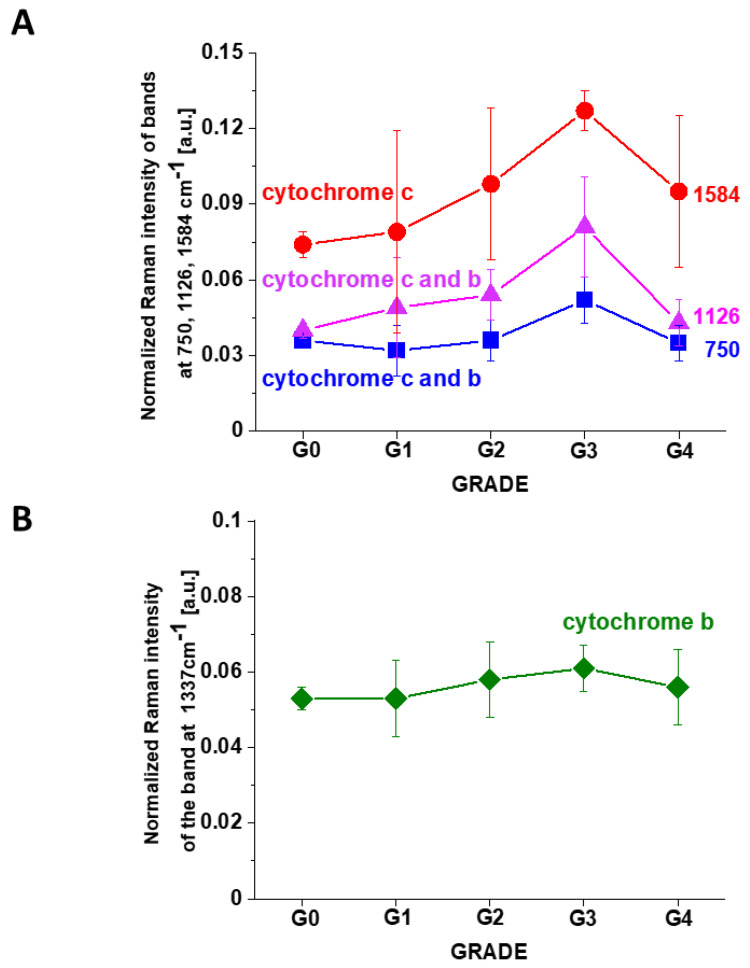
The Raman intensities of cytochrome *c* and cytochrome *b* in human brain tissue: I_750_, I_1126_, I_1584_ (**A**) and I_1337_ (**B**) as a function of brain tumor grade malignancy G0-G4 at excitation 532 nm. The results are presented as the mean ± SD. Raman bands intensity were taken from normalized by vector norm spectra.

## Data Availability

The raw data underlying the results presented in the study are available from Lodz University of Technology Institutional Data Access for researchers who meet the criteria for access to confidential data. The data contain potentially sensitive information. Request for access to those data should be addressed to the Head of Laboratory of Laser Molecular Spectroscopy, Institute of Applied Radiation Chemistry, Lodz University of Technology. Data requests might be sent by email to the secretary of the Institute of Applied Radiation Chemistry: mitr@mitr.p.lodz.pl.

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
