# Peer review of "Redox Imbalance and Biochemical Changes in Cancer by Probing Redox-Sensitive Mitochondrial Cytochromes in Label-Free Visible Resonance Raman Imaging"

_cancers, 2021, doi:10.3390/cancers13050960_

Round 1

Reviewer 1 Report

The authors have significantly improved the quality of this manuscript and I am happy to accept this version for publication.

Author Response

We would like to thank the Reviewer for positive opinion.

Reviewer 2 Report

The current research article entitled, "Redox Imbalance and Biochemical Changes in Cancer by Probing Redox-Sensitive Mitochondrial Cytochromes in Label-Free Visible Resonance Raman Imaging" is an exciting and far-reaching study showing mitochondrial redox status by localizing oxidative state of cytochromes C and hypothesized that Raman imaging can be used as a tool in the diagnosis of different grades of cancer. Once the authors have addressed the following specific comments this will be a significant contribution in the field.

Free electron transfer maintains intracellular redox balance. How did authors control/block the electron transfer upon excitation?

Oxidized cytochrome c (Fe3+) induces caspase whereas reduced form (Fe2+) cannot. The cytosolic cytochrome c reduced by various enzymes and/or reductants and prevent apoptosis, whereas, in apoptotic cells cytochrome rapidly oxidized by Mitochondrial Cyto. oxidase. The current manuscript didn't give any information on how is Raman imaging distinguishes the oxidized/reduced form of cytochrome c in mitochondria/cytoplasm transport?  Figure 5; Did the authors observe the effects of absorption spectra / Raman spectra on change in cytochrome C affinity with its reductant?

Line 244-246 and  Line 252 - 254; Figures 1, 2 and 4 ; authors need to combine the title of each figure with a footnote and add it below (footnote).

Figure 2B: define cts?

Methods: line 149: Authors need to define G0, G1, G2 and G3 in the method section in detail. 

Line 169; CaF2: provide catalog number and vendor. 

Line 177: provide complete make and vendor name of Alpha 300RSA.

Figure 4: provide staining protocol of Oil Red O staining protocol in the method section. 

Line 335 - 337, 363 - 366, 524: correct the symbol/character?

Author Response

Dear Reviewer, please find enclosed file.
